# Identification and Analysis of the *CBF* Gene Family in Three Species of *Acer* under Cold Stress

**DOI:** 10.3390/ijms24032088

**Published:** 2023-01-20

**Authors:** Qiushuang Zhao, Rui Han, Kewei Cai, Huiling Yan, Yan Li, Guanzheng Qu, Lin Liu, Xiyang Zhao

**Affiliations:** 1State Key Laboratory of Tree Genetics and Breeding, College of Forestry, Northeast Forestry University, Harbin 150040, China; 2Jilin Provincial Key Laboratory of Tree and Grass Genetics and Breeding, College of Forestry and Grassland Science, Jilin Agricultural University, Changchun 130118, China

**Keywords:** *Acer truncatum*, *Acer pseudosieboldianum*, *Acer yangbiense*, *CBF* gene family, cold stress

## Abstract

The C-Repeat Binding Factor (*CBF*) gene family has been identified and characterized in multiple plant species, and it plays a crucial role in responding to low temperatures. Presently, only a few studies on tree species demonstrate the mechanisms and potential functions of *CBFs* associated with cold resistance, while our study is a novel report on the multi-aspect differences of *CBFs* among three tree species, compared to previous studies. In this study, genome-wide identification and analysis of the *CBF* gene family in *Acer truncatum*, *Acer pseudosieboldianum*, and *Acer yangbiense* were performed. The results revealed that 16 *CBF* genes (five *ApseCBFs*, four *AcyanCBFs*, and seven *AtruCBFs*) were unevenly distributed across the chromosomes, and most *CBF* genes were mapped on chromosome 2 (Chr2) and chromosome 11 (Chr11). The analysis of phylogenetic relationships, gene structure, and conserved motif showed that 16 *CBF* genes could be clustered into three subgroups; they all contained Motif 1 and Motif 5, and most of them only spanned one exon. The cis-acting elements analysis showed that some *CBF* genes might be involved in hormone and abiotic stress responsiveness. In addition, *CBF* genes exhibited tissue expression specificity. High expressions of *ApseCBF1, ApseCBF3*, *AtruCBF1*, *AtruCBF4*, *AtruCBF6*, *AtruCBF7*, and *ApseCBF3*, *ApseCBF4*, *ApseCBF5* were detected on exposure to low temperature for 3 h and 24 h. Low expressions of *AtruCBF2*, *AtruCBF6*, *AtruCBF7* were detected under cold stress for 24 h, and *AtruCBF3* and *AtruCBF5* were always down-regulated under cold conditions. Taken together, comprehensive analysis will enhance our understanding of the potential functions of the *CBF* genes on cold resistance, thereby providing a reference for the introduction of *Acer* species in our country.

## 1. Introduction

Cold stress is a major natural factor that limits plant growth, productivity, and survival, and it determines the geographical distribution of plant species. Low temperature (0–12 °C) will inhibit plant growth and development, and freezing temperature (below 0 °C) will destroy cell membranes and cause cell death [1]. Plants can adapt to low temperatures through inducing the expression of cold tolerance-related genes, synthesizing corresponding protective proteins, and activating the protective enzymes and metabolites, and the process is called cold acclimation [2]. The cold acclimation natural habitat plants have the ability to acclimate completely, whereas plants originating from warmer climatic zones cannot often successfully overwinter, due to poor preparation for cold acclimation [3].

The genus *Acer* L. belongs to Aceraceae, which are deciduous or evergreen small trees or shrubs with medicinal, ornamental, and economic values. The ability of leaf color change determines the significant ornamental and economic values of *Acer*, and this ability is limited by diverse temperature zones. In Northeast China, some *Acer* species, such as *Acer pseudosieboldianum*, are mainly distributed in the cold temperate zone, and exhibit unique adaptability and cold resistance for tolerating temperatures below 30 °C in winter compared with other *Acer* species [4,5]. Most *Acer* species are native to Asia, whereas some species are spread through North America, Europe, and North Africa [6,7]. Over 150 species of *Acer* germplasm resources have primarily been found in China, accounting for more than half of the world’s *Acer* resources. The rich resources of germplasm in China are important for further research on the evolutionary history of the *Acer* species [8]. *Acer pseudosieboldianum*, *Acer yangbiense*, and *Acer truncatum* are three species of *Acer* endemic to China. *A. pseudosieboldianum* is endemic to north-east China [9]. *A. yangbiense* primarily grows in the western valley of Cangshan Mountain and Yunnan Province [10]. The natural distribution area of *A. truncatum* is mainly concentrated over north China and the north-east of China, and it is the most ubiquitous *Acer* species [11]. On the basis of distinct geographical environments and climates, various phenotypic differences and growth-specific differences of these three species are exhibited, and they may possess different cold acclimation.

As mentioned, cold acclimation is the inevitable result of a long period of low temperature time, affected by morphological coordination, physiological or biochemical adaptations, and genetic factors. The C-repeat binding factors dehydration-responsive element binding (CBFs/DREB1) proteins have been identified as key transcription factors involved in cold acclimation. They belong to the APETALA2/ethylene-responsive element binding factor (AP2/ERF) transcription factor family, which is further divided into the AP2, DREB, ERF, and Soloist, and related to abscisic acid insensitive 3/viviparous 1 (RAV) groups [12]. Among them, the DREB group consists of six subgroups, of which the A1 subgroup consists of the CBF/DREB1 transcription factors [13]. Studies have performed the genome-wide analysis of *CBF/DREB1* genes in several plant species, such as *Lolium perenne* [14], *Taraxacum kok-saghyz* [15], *Camellia sinensis* [16], *Eucalyptus grandis* [17], *Brassica rapa* [18] and so on. In previous studies, cold treatment (4 °C) induced the expression of *LpCBF3* in *Lolium perenne* [14], the *LeCBF1* gene in *Lycopersicon esculentum*, and the *CpCBF1* and *CpCBF2* in *Carica papaya* were also found to be cold-inducible [19,20]. In *Arabidopsis thaliana*, the *CBF* gene family contains six genes, including *CBF1*/*DREB1C*, *CBF2*/*DREB1B*, *CBF3*/*DREB1A*, *CBF4*/*DREB1D*, *DREB1E*/*DDF2*, and *DREB1F*/*DDF1* [21]. The expression of *AtCBF1*, *AtCBF2*, and *AtCBF3* is induced under cold stress, whereas the expression of *AtCBF4*, *AtDREB1E*, and *AtDREB1F* is induced under osmotic stress, such as drought and salt [22,23]. Studies have shown that *MbCBF2*, isolated from *Malus baccata*, enhances the resistance to cold stress in *A. thaliana* by increasing proline content, superoxide dismutase (SOD), peroxidase (POD), and catalase (CAT) activity [24]. The overexpression of *CsCBF1* increases the putrescine (Put) levels in Citrus sinensis, along with remarkably enhancing cold tolerance [25]. Besides low temperature, light quality, photoperiod, and the circadian clock also regulate the *CBF* gene expression via light-sensitive cis-elements in their promoter region [26]. 

In this study, we systematically performed genome-wide identification and analysis of the *CBF* gene family in *Acer pseudosieboldianum*, *Acer yangbiense*, and *Acer truncatum*, including comprehensive analysis of physical and chemical characteristics, chromosomal location, phylogenetic and evolutionary relationship, and conserved motifs. The expression pattern of the *CBF* genes in different tissues and the expression profile of three *Acer CBFs* under cold stress were detected. This was the novel report on the multi-aspect differences of *CBFs* among three tree species compared to previous studies. Our study will enhance our understanding of the *CBF* genes, which lay a theoretical foundation on the study of the CBFs protein structure and function, and the molecular breeding of resistance to cold in these three *Acer* species, thereby providing a reference for introducing *Acer* species in all regions in our country.

## 2. Results

### 2.1. Identification of CBF Genes in Three Acer Species

A total of 16 *CBF* genes were identified from three *Acer* species, including seven *AtruCBF* genes, four *AcyanCBF* genes, and five *ApseCBF* genes after sequence alignment with *A. thaliana*. Of these *CBFs*, *AtruCBF7* had the longest protein sequence with 395 amino acids, and the length of CDS was 1185 bp. The protein sequence of *AtruCBF5* was the smallest, with 165 amino acids, and the length of CDS was 495 bp. The molecular weight of CBF proteins ranged from 19,008.12 kDa (*AtruCBF5*) to 43,582.88 kDa (*AtruCBF7)*, and the isoelectric point ranged from 4.77 (*ApseCBF2*) to 9.79 (*AtruCBF4*). *ApseCBFs* and *AcyanCBFs* were neutral. The results of subcellular localization prediction revealed that most CBF proteins localized in the nucleus, while ApseCBF3, ApseCBF4, AcyanCBF1, AcyanCBF3, and AtruCBF5 proteins also localized in the cytoplasm concurrently (Table 1). These results suggested that significant differences existed in three *Acer* species *CBFs* genes.

### 2.2. Construction of Phylogenetic Tree

The phylogenetic relationships of the *CBF* gene family between three *Acer* species and other plants were analyzed. A total of 28 CBF proteins were used to construct the phylogenetic tree, including seven proteins from *A. truncatum* (*AtruCBFs*), four proteins from *A. yangbiense* (*AcyanCBFs*), five proteins from *A. pseudosieboldianum* (*ApseCBFs*), six proteins *A. thaliana* (*AtCBFs*), and six proteins from *P. trichocarpa* (*PtrCBFs)*. These *CBFs* were clustered in five groups: Group I, Group II, Group III, Group IV, and Group V. The CBFs from three *Acer* species were clustered in Group I, Group IV, and Group V. Group I was the largest group, containing one *AcyanCBFs*, one *ApseCBFs*, five *AtCBFs, AtDDF1*, and *AtDDF2*. Four *PtrCBFs* were clustered in Group III. Interestingly, all five species are clustered to group V (Figure 1). These results suggested that the evolution relationships of CBFs in *Acer* species were different with other plant species.

### 2.3. Gene Structure and Conserved Motif of CBF Genes

The structure and motif of *CBF* genes in *A. truncatum*, *A. pseudosieboldianum*, and *A. yangbiense* were analyzed, and these 16 *CBFs* were ordered according to the phylogenetic tree (Figure 2a). A total of 12 *CBFs* spanned only one exon, and four *CBFs*. *AcyanCBF1*, *AcyanCBF2*, *AtruCBF2*, and *AtruCBF6* spanned two exons and one intron (Figure 2c). The analysis of conserved motifs showed that the *CBF* genes had three to eight motifs. Motif 1 and Motif 5 were found from all *CBFs*. Motif 2 and Motif 4 were found from *AcyanCBFs* and *ApseCBFs*, while Motif 6 was only found from *AtruCBFs*. Further, Motif 3 was found in *AcyanCBFs* and *ApseCBFs*, except for *ApseCBF4* and *AcyanCBF3*. Motif 7 was only found in *AcyanCBF1*, *AcyanCBF2*, *ApseCBF3*, *ApseCBF1*, and *ApseCBF2*. Motif 10 was only found in *AtruCBF1*, *AtruCBF4*, *AtruCBF5*, *AtruCBF6*, and *AtruCBF7* (Figure 2b). These results suggested that the gene structure and motif of *CBFs* in three *Acer* species were relatively conserved.

### 2.4. Cis-Acting Elements of CBF Genes

The cis-acting elements of the 16 *CBF* genes promoters were explored, and a total of 24 types of cis-elements were identified. Low-temperature response elements were found in the promoter of three *ApseCBFs* genes, including *ApseCBF3*, *ApseCBF4*, and *ApseCBF5*; three *AcyanCBFs* genes, including *AcyanCBF1*, *AcyanCBF2*, *AcyanCBF3*; and two *AtruCBFs* genes, including *AtruCBF4* and *AtruCBF7*. The light responsiveness presented in the promoter of all *CBF* genes with a large number. The promoter of *CBF* genes also contained elements associated with hormone responsiveness, such as MeJA-responsiveness elements, found in the promoter of *ApseCBFs*, and gibberellin-responsiveness elements, found in the promoter of *ApseCBF3* and *AtruCBF6*. The promoter of *ApseCBF* genes had drought inducibility elements, but of all *AtruCBFs* genes, the drought inducibility elements only presented in the promoter of *AtruCBF5* (Figure 3). The detection of these cis-acting elements suggested that the *CBF* genes in three *Acer* species played important roles in treating abiotic stress, drought, light and cold stress, etc.

### 2.5. Chromosome Location of CBF Genes

The chromosomal location of *CBF* genes in three *Acer* species were analyzed. The *ApseCBFs* genes were unevenly distributed at one end of three chromosomes in *A. pseudosieboldianum*, where *ApseCBF1*, *ApseCBF2*, *ApseCBF4* were located on Chr2, *ApseCBF5* was located on Chr7, and *ApseCBF3* was located on Chr11 (Figure 4a). The *AcyanCBF* genes were also unevenly distributed at one end of three chromosomes in *A. yangbiense*. *AcyanCBF3* was located on Chr1, *AcyanCBF4* was located on Chr2, and *AcyanCBF1* and *AcyanCBF2* were located on Chr11 (Figure 4b). The *AtruCBF* genes dispersed six chromosomes, and each *AtruCBF* gene was located on one chromosome, except for Chr4 containing *AtruCBF4* and *AtruCBF7*. *AtruCBF2*, *AtruCBF3*, *AtruCBF6*, *AtruCBF1*, and *AtruCBF4* were located on Chr1, Chr2, Chr6, Chr10, and Chr13, respectively (Figure 4c). The *CBF* gene family was not located on all 13 chromosomes; this difference in gene distribution determined the complexity and diversification of *CBFs* in three *Acer* species, providing clues to their evolution.

### 2.6. Synteny Analysis of CBF Genes

The intraspecific and interspecific synteny analysis of *CBF* genes were explored. One orthologous gene pair located on Chr2 and Chr11 was found in *A. pseudosieboldianum* and *A. yangbiense* (Figure 5a,b). Two orthologous gene pairs located on Chr3 and Chr13, Chr6 and Chr12 were found in *A. truncatum* (Figure 5c). As for the synteny analysis between different species, *A. yangbiense* and *A. truncatum*, *A. truncatum* and *A. pseudosieboldianum* contained 12 and 14 orthologous gene pairs, indicating they possessed the higher homology. *A. yangbiense* and *A. pseudosieboldianum* only contained five orthologous gene pairs, indicating they possessed lowest homology. The results showed that *A. pseudosieboldianum* and *A. truncatum* exhibited the highest level of homology. Furthermore, orthologous gene pairs in *A. pseudosieboldianum* and *A. truncatum* were mainly distributed on chromosomes 2 and 1. Further, some orthologous gene pairs were also detected between *A.thaliana* and *A. pseudosieboldianum*, *A.thaliana* and *A. yangbiense*, and *A. thaliana* and *A*. *truncatum*, suggesting these species also exhibited homology (Figure 5d). These results proposed that *CBF* genes possessed a degree of homology in different *Acer* species. 

### 2.7. The Expression of CBF Genes in Different Tissues

The expression of *CBF* genes in different tissues was explored based on the transcriptomes. In *A. pseudosieboldianum*, a higher expression of *ApseCBFs* genes was detected in green leaf (GL) compared to half-red leaf (HRL) and red leaf (RL) (Figure 6a). In *A. yangbiense*, the expressions of *AcyanCBF1*, *AcyanCBF2*, and *AcyanCBF3* were up-regulated in the stem and sprout, whereas these genes were down-regulated in the leaf and fruit (Figure 6b). In *A. truncatum*, the expressions of *AtruCBF5* and *AtruCBF6* were up-regulated in the flower and seed, while the expressions of *AtruCBF3*, *AtruCBF4*, and *AtruCBF1* were up-regulated in seed, flower, and root, respectively (Figure 6c). The results revealed that the *CBF* genes possessed tissue-specific expression. 

### 2.8. Expression of CBF Genes in Three Acer Species under Low-Temperature Conditions

Previous studies have shown that *CBF* genes play an important role in responding to cold stress [24,25]. Therefore, the expression of *CBF* genes in three *Acer* species under low temperatures was detected. The results showed that low temperatures could induce the expression of *CBF* genes (Figure 7). The expression of *ApseCBF1*, *ApseCBF3*, *AtruCBF1*, *AtruCBF4*, *AtruCBF6*, and *AtruCBF7* was up-regulated on exposure to low temperature for 3 h, and the differences were significant except for *ApseCBF1* and *AtruCBF7* (*p* < 0.05) (Figure 7a,c,f,i,k,l). *ApseCBF3*, *ApseCBF4*, and *ApseCBF5* were significantly highly expressed under cold for 24 h (*p* < 0.05) (Figure 7b,d,e), while *AtruCBF2*, *AtruCBF6*, and *AtruCBF7* were significantly lowly expressed at 24 h (*p* < 0.05) (Figure 7g,k,l). Moreover, the expressions of *AtruCBF3* and *AtruCBF5* were always down-regulated during the low temperature conditions (Figure 7h,j). The expressions of *AcyanCBF1* and *AcyanCBF2* exhibited no significant differences under cold conditions, except for at 3 h (Figure 7m,n), while a significantly higher expression of *AcyanCBF3* was detected on exposure to low temperature for 12 h and 48 h (Figure 7o). These results suggested that *CBF* genes in three *Acer* species have specific expression patterns when exposed to low-temperature conditions for different time durations.

## 3. Discussion

*Acer* is a medicinal, ornamental, and economic tree species. The *Acer* species mentioned in our study, *A. pseudosieboldianum*, *A. yangbiense*, and *A. truncatum*, are distributed across different areas of China. Presently, the studies on genome assembly of *A. pseudosieboldianum*, *A. yangbiense*, and *A. truncatum* heve been reported, which provide references for exploring the genome-wide identifications and functions of genes in three species [8,27,28]. CBF transcription factors, also known as DREB1 proteins, belong to the AP2/ERF family. They play an important role in mediating plant responses to biotic and abiotic stress, including high salt, drought, and low temperature. Particularly, *CBFs* can enhance the resistance to cold stress in several tree species, such as *Eucalyptus gunnii*, *Malus* × *domestica*, and *Betula pendula* [29,30,31]. On the basis of their genome, our study performed genome-wide identification and analysis of *CBFs* in three *Acer* species, which will lay the foundation for further study of *CBF* gene functions in cold resistance. 

Five, four, and seven *CBF* genes were identified from *A. pseudosieboldianum*, *A. yangbiense*, and *A. truncatum*. Most CBF proteins are localized predominantly in the nucleus (Table 1), in accordance with their biological roles as transcription factors. The study on identifying the gene family based on the whole genome is significant for understanding the origin, evolution, and differentiation of gene families [32]. These results were similar to the report on *CsCBFs* [16]. According to the phylogenetic tree analysis, the results showed that the *CBF* family members of the three species of *Acer*, *P. trichocarpa*, and *A. thaliana* did not form a separate cluster of evolutionary branches, indicating that there was a certain degree of homology between the *CBF* gene families of several species. At the same time, the *CBF* gene family may have undergone great differentiation in evolution (Figure 1) [15]. Although *CBF* homologs from *A. thaliana*, *A. pseudosieboldianum*, and *A. yangbiense* were clustered in the same group, they were divided into separate subgroups, indicating that the duplications of *CBF*s in eudicot plants were independent events, and the duplication and divergence occurred after speciation [33]. The analysis of gene structure and conserved motif showed that genes clustered on the same branch exhibited similarity, such as similar exon number, motif number, and motif position (Figure 2a–c). The certain conserved motifs played important functional and/or structural roles in active proteins [34]. All *ApseCBFs*, most *AcyanCBFs*, and *AtruCBFs* were single-exon structures, while *AcyanCBF1*, *AcyanCBF2*, *AtruCBF2*, *AtruCBF6* contained one intron (Figure 2c). The results showed that most members of the CBF family in three *Acer* species were intron deletion genes, which were consistent with *CBFs* in *Taraxacum koksaghyz* [15]. The loss of introns might shorten the time required for gene transcription to translation, thereby accelerating gene expression and functional protein production to adapt to the changes of the plant and environment [35]. 

Most *ApseCBFs*, *AcyanCBFs*, and *AtruCBFs* were unevenly distributed at one end of the chromosomes in three Acer species, while *ApseCBF1*/2/4, *AcyanCBF1*/2, *AtruCBF4*/7 were located on one chromosome (Figure 4a–c). The difference in gene distribution determined the complexity and diversification of *CBFs* in three *Acer* species, which might be caused by the differences in the structure and size of the chromosomes. The intraspecific and interspecific synteny analysis of *CBFs* could also be used to indicate the homology. Interestingly, *A. yangbiense* and *A. truncatum*, *A. truncatum* and *A. pseudosieboldianum* had higher homology, while *A. yangbiense* and *A. pseudosieboldianum* had lower homology (Figure 5d). The distribution areas of *A. yangbiense* and *A. pseudosieboldianum* were distinct; one was distributed in Northeast China, the other was distributed in Yunnan Province [9,10]. The regional differences might lead to the separation of plants and gene origin, divergence, and evolution [36]. 

The gene promoters are upstream of the transcriptional start, which contains plenty of cis-acting elements, and controls the transcription of genes [37]. The promoter polymorphisms of *CBFs* affect the expression of *CBF* genes, and affect the expression of related response genes in *A. thaliana* [38,39,40]. The studies have also shown that *AtCBF2* negatively regulated *AtCBF3* and *AtCBF1*, while *AtCBF4* functioned in drought stress tolerance [22,41]. In our study, low-temperature, drought, light, and plant hormone-responsive elements were identified from the promoter region of *ApseCBFs*, *AcyanCBFs*, and *AtruCBFs* genes. For example, the low-temperature responsiveness cis-acting elements were found in the promoter of *ApseCBF4*, *ApseCBF5*, *ApseCBF3*, *AcyanCBF1*, *AcyanCBF2*, *AtruCBF4*, *AtruCBF7*, and the expression of these genes could be significantly induced by cold stress, especially *ApseCBF4* at 12 h, *ApseCBF5* at 24 h, *ApseCBF3* at 3 h, *AcyanCBF1* at 6 h, *AcyanCBF2* at 12 h, *AtruCBF4* at 3 h, *AtruCBF7* at 3 h (Figure 3 and Figure 7). It is speculated that some transcription activators could specifically bind to and activate the promoters of *CBFs* on exposure to low temperature, thereby inducing the expression of these mentioned genes; therefore they may play crucial roles in cold resistance [14,42]. Other cis-acting elements, such as drought-inducibility elements in the promoters of *ApseCBFs*, *AcyanCBF2*/4, *AtruCBF5*; defense and stress responsiveness; and wound-responsive elements were found (Figure 3). These elements might help in activating *CBFs* to cope with other abiotic stresses. The expression patterns of *CBFs* under cold stress were explored. A high expression of some *CBF* genes on exposure to low temperature for 3 to 12 h was detected, while an increase in expression of *ApseCBF4*, *AcyanCBF3*, and *AtruCFB2* on exposure to low temperature for 12 h, *AcyanCBF2*, and *AcyanCBF4* on exposure to low temperature for 12 to 48 h was observed (Figure 7). The results indicated that some *CBF* genes might function during the early stages of response to cold stress, and that genes such as *AcyanCFB2*/3, and *AcyanCBF4* might function during the late stages of response to cold stress. Previous studies have also demonstrated that the *CBF* genes respond to cold stress in a time-dependent manner in *Secale cereale* L. [43] and *Camellia sinensis* [16]. However, no significantly negative regulations were found in three *Acer* species between *CBF2* and *CBF1*/3 from gene expression patterns under low temperature, so we will explore this relationship in the future. Further, the tissue-specific expression of *CBF* genes was exhibited (Figure 6a–c); these results were also reported in *Eucalyptus grandis* and *Punica granatum* [17,44]. In summary, our study excavated some *CBFs* significantly induced by low temperature in three *Acer* species, which provided a reference for further gene function research and molecular regulation mechanisms in cold resistance. The results might direct the introduction of *Acer* species.

## 4. Materials and Methods

### 4.1. Plant Materials

The three-year-old *A. truncatum*, *A. yangbiense* and *A. pseudosieboldianum* were obtained from the Northeast Forestry University greenhouse (126°38′8.92″ E, 45°43′20.64″ N), Harbin, Heilongjiang Province, China. *Acer* species were exposed to low temperatures (i.e., 4 °C) in an intelligent light incubator (ToppYunnong Technology Co., Ltd., Hangzhou, China) for 0 h, 3 h, 6 h, 12 h, 24 h, and 48 h. The functional leaves (the third to fifth leaves from the main branches) were collected, and total RNA was extracted to analyze the expression pattern of *A. pseudosieboldianum CBFs* (*ApseCBFs*), *A. yangbiense CBFs* (*AcyanCBFs*), and *A. truncatum CBFs* (*AtruCBFs*) genes.

### 4.2. Retrieving the CBF Gene Family Sequences

To perform genome-wide analysis of the *CBF* genes from three *Acer* species, the whole genome sequences were directly obtained according to previous studies [8,27,28]. The six known CBF transcription factor family genes from *A. thaliana* were selected as the query objects, and the protein sequences were retrieved using *the Arabidopsis* Information Resource (TAIR) (https://www.arabidopsis.org/browse/genefamily/index.jsp, accessed on 12 December 2021) [45]. The CBF genes in *A. truncatum*, *A. yangbiense* and *A. pseudosieboldianum* were identified using the BLAST by Toolbox for Biologists (TBtools) v 1.087 (e-value < 1 × 10^−5^) based on *A. thaliana* [46]. Each *A. thaliana* gene was successfully matched with multiple *CBF* genes, and the alignment sequence IDs of candidate *CBFs* were obtained eventually after eliminating the repeated values and blanks. The candidate *CBF* genes were further manually analyzed using Batch CD-Search in the National Centre for Biotechnology database (NCBI) (https://www.ncbi.nlm.nih.gov/Structure/bwrpsb/bwrpsb.cgi, accessed on 12 December 2021) to detect the presence of the *CBF* domain, and 16 candidate *CBFs* were identified. The biochemical properties, such as the molecular weight (MW) and isoelectric point (pI), were determined using the Compute pI/Mw tool on the ExPASy (https://web.expasy.org/protparam, accessed on 15 December 2021). The subcellular localization was analyzed using the WoLF PSORT tool (https://wolfpsort.hgc.jp/, accessed on 15 December 2021). 

### 4.3. Analyses of Gene Structure and Protein Motif Composition

The structures of candidate *CBF* genes in *A. truncatum*, *A. pseudosieboldianum*, and *A. yangbiense* were identified and visualized using the TBtools software [47]. MEME, a web-based tool (http://meme-suite.org/tools/meme, accessed on 20 December 2021), was used to explore the conserved motif in *A. truncatum*, *A. pseudosieboldianum*, and *A. yangbiense*. The parameters were set at a maximum of 10 motifs. TBtools was further used to visualize the motif composition [48]. 

### 4.4. The Analyses of Chromosomal Location and Collinearity

The chromosome locations of *CBFs* in three *Acer* species were analyzed, and they were mapped on 13 chromosomes (named Chr 1 to Chr 13) according to their physical positions (bp). The McScan software was used to perform the collinearity analysis, and homology between *ApseCBFs*, *AcyanCBFs*, *AtruCBFs* and *AtCBFs* was evaluated with default parameters. These results were visualized using TBtools.

### 4.5. The Analysis of Phylogenetic Tree

The evolutionary relationship of *CBF* genes in *A. thaliana*, *A. pseudosieboldianum*, *A. yangbiense*, *A. truncatum*, and *Populus trichocarpa* was analyzed. The phylogenetic tree was constructed using the Neighbor-joining method by MEGA software, the Bootstrap method value was set to 1000, and other parameters were set to default. Interactive Tree of Life (iTOL) (https://itol.embl.de/, accessed on 22 December 2021) was used to beautify the phylogenetic tree.

### 4.6. The Analysis of Cis-Acting Elements

The 2000 bp sequences upstream of *CBF* genes coding sequences (CDSs) were extracted as promoter sequences using TBtools. The cis-acting elements were identified using the Plant CARE (http://bioinformatics.psb.ugent.be/webtools/plantcare/html). 

### 4.7. The Analysis of Gene Expression

The RNA-seq data of *A. pseudosieboldianum*, *A. yangbiense*, and *A. truncatum* were obtained from NCBI (accession numbers were PRJNA736515, PRJNA524417, and PRJNA665613), respectively. After quality control, alignment, and quantitative analysis, the expression levels of *CBF* genes were represented using fragment per kilobase per million mapped reads (FPKM). The gene expression patterns of green leaf (GL), half-red leaf (HRL), and red leaf (RL) in *A. pseudosieboldianum*; sprout, leaf, root, fruit, and stem in *A. yangbiense*; and flower, leaf, root, seed, stem in *A. truncatum* were explored.

Total RNA was extracted from the functional leaves of three *Acer* species under cold stress (4 °C for 0, 3, 6, 12, 24, and 48 h) using RNAprep Pure Plant Kit (Tiangen Biotech, Beijing, China) according to the manufacturer’s instructions. After detecting the integrity and quality of total RNA, the cDNA was synthesized using PrimeScriptTM RT reagent Kit with gDNA Eraser (TaKaRa, Beijing, China). A total of 16 *CBF* genes were selected to explore the gene expression using qRT-PCR. The primer sequences were listed in Appendix A, and *18S* was used as an internal reference gene. The qRT-PCR was performed on the ABI 7500 Real-Time PCR system (Applied Biosystems, Carlsbad, CA, USA), using TB Green^®^ Premix Ex Taq^TM^ II (Tli RNaseH Plus) (TaKaRa, Beijing, China) with three technical replicates. Amplification system and procedure were carried out according to Li [32]. The relative gene expression levels were calculated using the 2^−ΔΔCT^ method [49].

## 5. Conclusions

In conclusion, we have identified seven *AtruCBF* genes, four *AcyanCBF* genes, and five *ApseCBF* genes from *A. truncatum, A. yangbiens*, and *A. pseudosieboldianum*. These *CBF* genes, clustered in five subgroups based on phylogenetic relationships, mainly contained conserved Motif 1 and Motif 5, and 12 *CBF* genes only spanned one exon. The cis-acting elements in the promoters of *CBF* genes were involved in hormone, light, drought, and low-temperature responsiveness. The *CBFs* were unevenly distributed at chromosomes in three *Acer* species, mostly at Chr2 and Chr11. One, one, and two orthologous gene pairs were found in *A. pseudosieboldianum, A. yangbiense*, and *A. truncatum. A. yangbiense* and *A. truncatum, A. truncatum* and *A. pseudosieboldianum* possessed high homology, while *A. yangbiense* and *A. pseudosieboldianum* possessed the lowest homology. In addition, high expressions of *ApseCBF1*, *ApseCBF3*, *AtruCBF1*, *AtruCBF4*, *AtruCBF6*, *AtruCBF7*, and *ApseCBF3*, *ApseCBF4*, *ApseCBF5* were detected on exposure to low temperature for 3 h and 24 h. Low expressions of *AtruCBF2*, *AtruCBF6*, *AtruCBF7* were detected under cold stress for 24 h, and *AtruCBF3* and *AtruCBF5* were always down-regulated under cold conditions. These results provided a meaningful direction for gene function research on the cold-resistance of *A. truncatum*, *A. pseudosieboldianum*, and *A. yangbiens*, which is favorable for the future introduction of these three *Acer* species. 

## Figures and Tables

**Figure 1 ijms-24-02088-f001:**
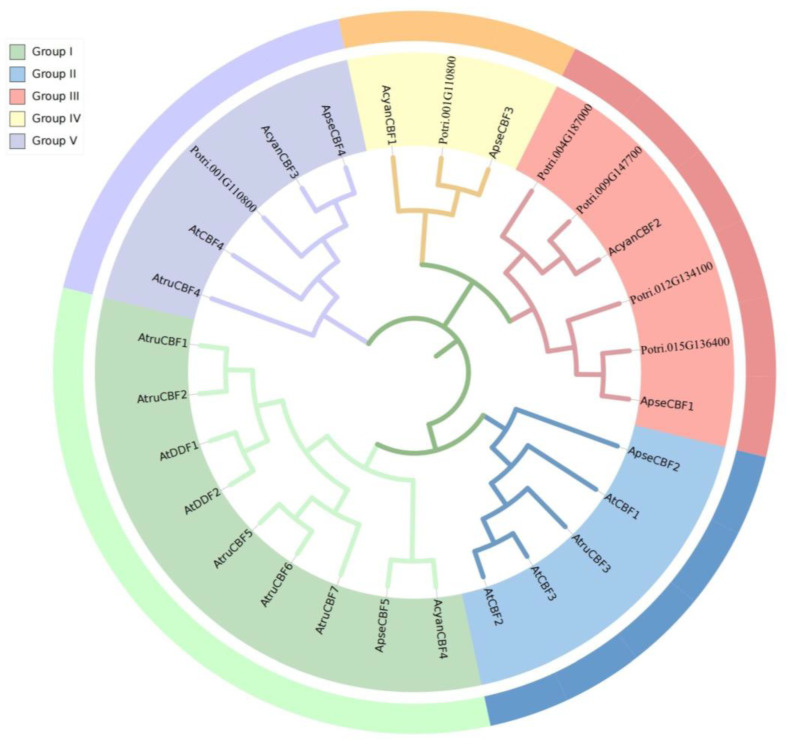
Phylogenetic tree of the CBF proteins from six species. Different colors represent the groups. At: *A. thaliana*; Acyan: *A. yangbiense*; Atru: *A. truncatum*; Apse: *A. pseudosieboldianum*; Os: *O. sativa*; Potri: *P. trichocarpa.*

**Figure 2 ijms-24-02088-f002:**
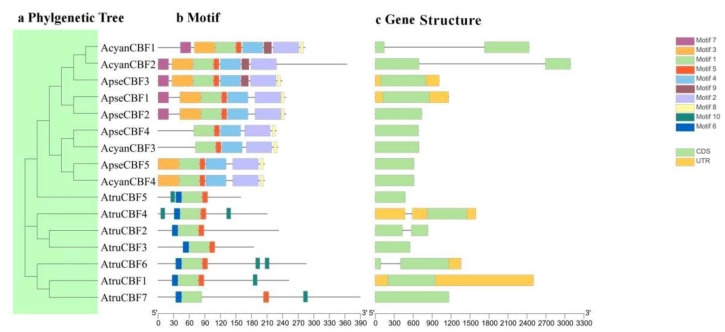
The analysis of gene structure and conserved motif. (**a**) The phylogenetic tree of all CBF proteins in three *Acer* species. (**b**) The motif composition of *CBF* genes in three *Acer* species. (**c**) The structure of *CBF* genes.

**Figure 3 ijms-24-02088-f003:**
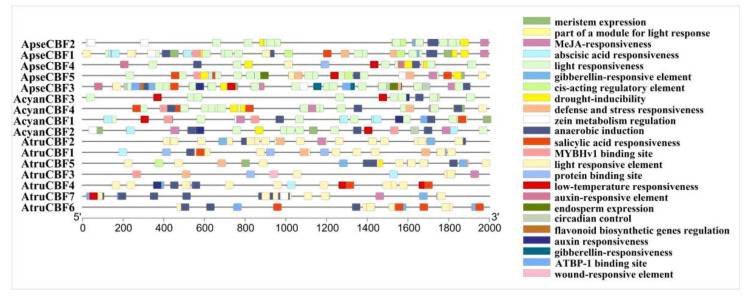
The number and position of cis-acting elements in three *Acer* species *CBFs* promoters. Different color boxes represent different elements.

**Figure 4 ijms-24-02088-f004:**
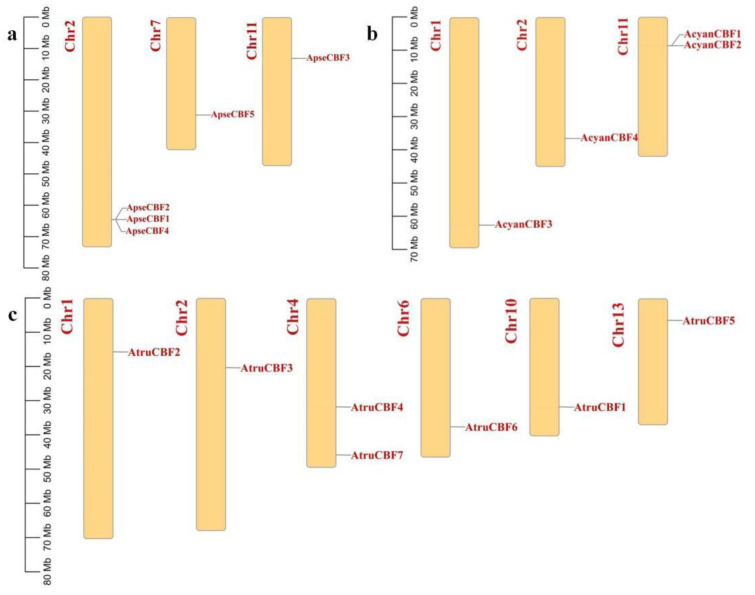
(**a**) The chromosomal location of *ApseCBFs* in *A*. *pseudosieboldianum*. (**b**) The chromosomal location of *AcyanCBFs* in *A. yangbiense*. (**c**) The chromosomal location of *AtruCBFs* in *A. truncatum*.

**Figure 5 ijms-24-02088-f005:**
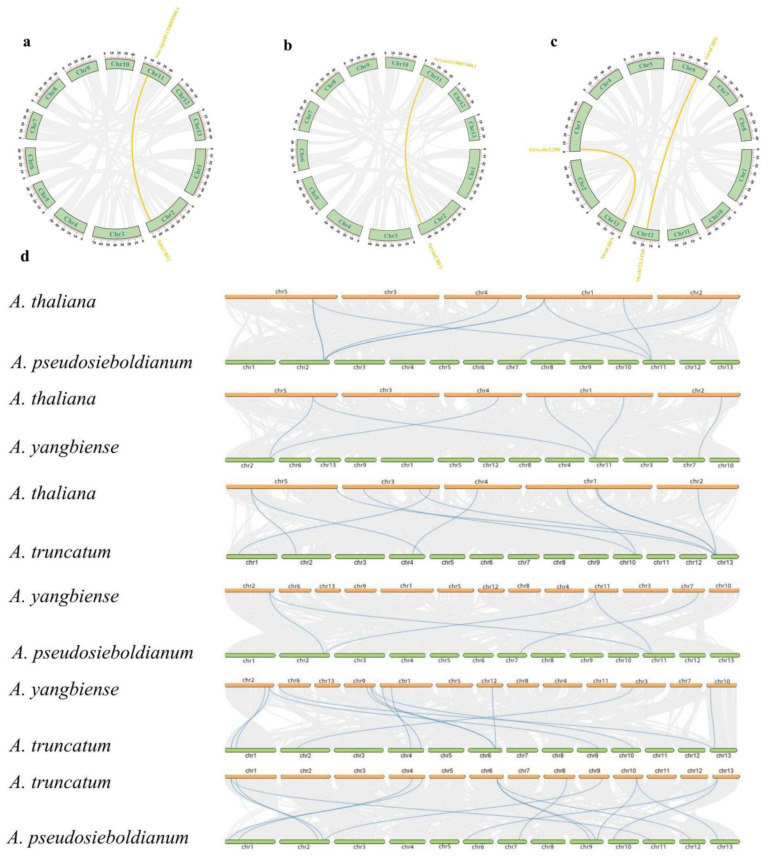
(**a**) The synteny of *ApseCBFs* in *A. pseudosieboldianum*. (**b**) The synteny of *AcyanCBFs* in *A. yangbiense*. (**c**) The synteny of *AtruCBFs* in *A. truncatum*. (**d**) The synteny analysis of *CBFs* between *A. thaliana* and *A. pseudosieboldianum*, *A. thaliana* and *A. yangbiense*, *A. thaliana* and *A. truncatum*, *A. yangbiense* and *A. pseudosieboldianum*, *A*. *yangbiense* and *A. truncatum*, and *A. truncatum* and *A. pseudosieboldianum*. The orthologous gene pairs are highlighted using blue lines.

**Figure 6 ijms-24-02088-f006:**
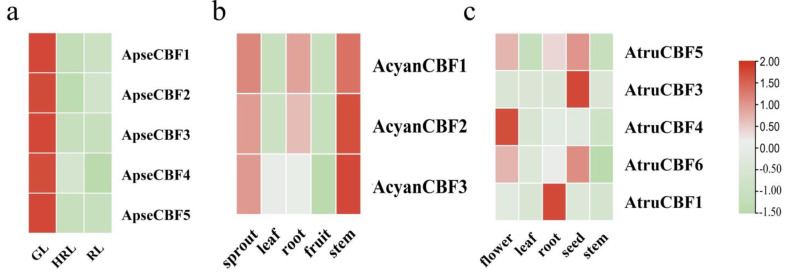
The expression pattern of the *CBF* genes in different tissues. (**a**) The expression of five *ApseCBF* genes in green leaf (GL), half-red leaf (HRL), red leaf (RL) of *A. pseudosieboldianum*. (**b**) The expression of three *AcyanCBF* genes in sprout, leaf, root, fruit, and stem of *A. yangbiense*. (**c**) The expression of five *AtruCB*F genes in flower, leaf, root, seed, and stem of *A. truncatum*.

**Figure 7 ijms-24-02088-f007:**
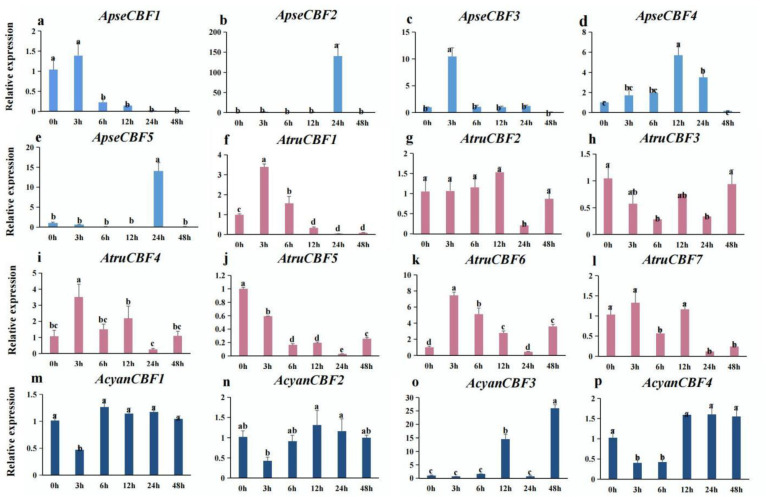
(**a**–**e**) The expression of *CBF* genes in *A. pseudosieboldianum* under cold stress. (**f**–**l**) The expression of *CBF* genes in *A. truncatum* under cold stress. (**m**–**p**) The expression of *CBF* genes in *A. yangbiense* under cold stress. The *Y*-axis and *X*-axis represent the gene expression level detected by qRT-PCR and treatment time, respectively. The values are the mean ± SD of three technical replicates. The letter indicates significant differences (*p* < 0.05).

**Table 1 ijms-24-02088-t001:** The identification of CBF genes in three Acer species.

Gene Name	Gene ID	AA ^1^	MW ^2^ (kDa)	PI ^3^	SL ^4^	CDS (bp)
*ApseCBF1*	Apse002T0243400.1	253	27,693.81	4.9	Nucleus	759
*ApseCBF2*	Apse002T0243300.1	253	27,666.73	4.77	Nucleus	759
*ApseCBF3*	Apse011T0099600.1	246	27,542.91	6.68	Cytoplasm, Nucleus	738
*ApseCBF4*	Apse002T0243600.1	235	24,831.5	5.04	Cytoplasm, Nucleus	705
*ApseCBF5*	Apse007T0094500.1	212	22,876.3	5.46	Nucleus	636
*AcyanCBF1*	Acyan11G0087800.1	301	33,234.08	6.03	Cytoplasm, Nucleus	903
*AcyanCBF2*	Acyan11G0087900.1	382	42,556.73	8.9	Cytoplasm	1146
*AcyanCBF3*	Acyan02G0289000.1	249	26,360.25	5.42	Cytoplasm, Nucleus	747
*AcyanCBF4*	Acyan07G0093600.1	223	23,930.49	5.61	Nucleus	669
*AtruCBF1*	Atru.chr10.1810	258	27,824.85	9.35	Nucleus	774
*AtruCBF2*	Atru.chr1.1042	238	27,303.82	9.49	Nucleus	714
*AtruCBF3*	Atru.chr2.1230	190	20,944.53	9.73	Nucleus	570
*AtruCBF4*	Atru.chr4.1701	216	24,514.44	9.79	Nucleus	648
*AtruCBF5*	Atru.chr13.731	165	19,008.12	9.37	Cytoplasm, Nucleus	495
*AtruCBF6*	Atru.chr6.2697	291	31,912.79	7.37	Nucleus	873
*AtruCBF7*	Atru.chr4.2790	395	43,582.88	6.38	Nucleus	1185

^1^ The number of amino acids, ^2^ Molecular weight, ^3^ Isoelectric point, ^4^ Subcellular localization.

## Data Availability

Data are contained within the article and Appendix A.

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
