# Peer review of "Identification and Analysis of the CBF Gene Family in Three Species of Acer under Cold Stress"

_ijms, 2023, doi:10.3390/ijms24032088_

Round 1

Reviewer 1 Report

The manuscript «Identification and analysis of the CBF gene family in three species of Acer under cold stress» of Zhao Q. et al. described the structure and identification of CBF gene family in three tree’s species.

In this manuscript well described the localization and phylogenetic relationship between genes of CBF family. The work was made using modern technology and methods. Commonly the manuscript seems interesting but some questions and remarks listed below:

1.      The role of CBF genes in cold resistance well known. But CBF is family of transcription factors that are regulatory proteins and they induced not only by cold. CBF factors play plural role in stress tolerance and induced by different stressors. Perhaps in this case it is better to investigate the target genes (for example COR, LEA etc., some of them can be specific, such as WCS 120 in wheat) for further analysis of species for introduction. For avoid this, maybe better in originality add information about: Is it firstly described CBF gene in Acer? If it is not, in which trees CBF were found?

2.      Why for phylogenetic tree construction was chosen several species? Is any information about others trees not about monocot (rice)?

3.      On the Fig 7 there no OY-axis description. Is it units? Transcript level? Also it will be better added statistic analysis, because in some graphs (g, h) the units such as 0.0001, 0.0002 seems less detection capability.

4.      The list of references must be prepared according to instruction of journal

Reviewer 2 Report

1. The description of medicinal and ornamental value of Acer L. was not directly related to its cold adaptation. The literature review should focus on what is the significance of studying Acer L. on cold resistance? Scientific value or plantation development value

2.  In introduction, there were few descriptions about the response of CBF gene family members of different species to cold stress.

3. In line 102-104, the molecular weight of physical and chemical properties ranges from 222876.3 to 43582.88kDa, and the isoelectric point ranges from 4.9-9.49. The author should have made a mistake in the number, and it is suggested to check and modify it again. In addition, the medium electric point in the figure is basically greater than 5, and ApseCBFs and AcyanCBFs should be considered neutral. Rather than weakly acidic, AtruCBFs have several isoelectric points in the neutral range, so it should not be generalized to classify AtruCBFs as weakly alkaline.

4. The subcellular localization prediction is different from the subcellular localization experiment. In this paper, only the subcellular localization prediction was made. Therefore, there were expression errors. In addition, in 2.2, at least two methods should be used to predict subcellular localization to improve accuracy.

5. Why was the species O. sativa (OsCBFs) added to the phylogenetic tree in 2.2 of line 2,113 for comparison? This species was not discussed in the paper. In the phylogenetic tree constructed in Figure 1, this species was also separately divided and did not share homology with the three Acer species discussed in the paper. So I think maybe taking the sequence out of the tree doesn't make much difference to the content.

6. The motif analysis content in line 3,362 says that the maximum parameter used in meme online software is set to 8, but there are 10 motif motif motif in Figure2, please check and correct, and whether the motif serial number can be marked in the figure, motif5 and motif9 are not easy to identify. Makes me think Motif 1 and Motif 5 were found from all CBFs. This sentence is up for debate.

7. The light responsiveness elements presented in the promoter of all CBF genes with large number, The light responsiveness elements presented in the promoter of all CBF genes with large number, light responsiveness elements are not found in AcyanCBFs and ApseCBFs according to the cis-acting elements reflected in Figure3.

8. and they were mapped on 13 chromosomes (named Chr 1 to Chr 13) according to their physical positions (bp), and they were mapped on 13 chromosomes (named CHR 1 to CHR 13) according to their physical positions (BP), As shown in Figure 4, the CBF gene family is not located on all 13 chromosomes. Meanwhile, The difference of gene distribution determined the complexity and diversification of CBFs in lines 282 to 284 three Acer species, which might be caused by the differences in structure and size of the I feel that this sentence may not be speaking in the correct way (I have not read such a phrase in such a paper).

9. The title of line 6,186 is collinear analysis, but in this content, the author basically just described the situation in the picture without making any analytical expression, so the content may need to be supplemented.

10. in line 7,205, the Expression of CBF Genes in Different Tissues is why the other two genes are expressed in different tissues, while apseCBFs is an expression map constructed by the regulation of different colors of light. Perhaps the author needs to add to this or make an explanation.

11. Some contents of the Discussion could be modified to the Result.

12. The analysis and explanation of the significant difference between the expression of different cold stress were lacking in Figtrue7. The author explained the quantitative results in 2.8 but did not explain the expression rules.

13. The author useed 18S as the internal reference gene, but it was not shown in the supplementary. Why you chose 18S?

Round 2

Reviewer 1 Report

The recomendations were accepted and adequate answers were given.  The manuscript can be recommend for publishing

Reviewer 2 Report

The author has effectively modified the manuscript, and I think the standard now is can be published.